# The combined tumour-based Fascin/Snail and stromal periostin reveals the effective prognosis prediction in colorectal cancer patients

Niphat Jirapongwattana[1]☯, Suyanee Thongchot[1,2]☯, Ananya Pongpaibul[3], Atthaphorn Trakarnsanga[4], Jean Quinn[5], Peti Thuwajit[1], Chanitra Thuwajit[1]*, Joanne Edwards[5]*

1 Department of Immunology, Faculty of Medicine Siriraj Hospital, Mahidol University, Bangkok, Thailand, 2 Research Department, Siriraj Center of Research Excellence for Cancer Immunotherapy (SiCORE-CIT), Faculty of Medicine Siriraj Hospital, Mahidol University, Bangkok, Thailand, 3 Department of Pathology, Faculty of Medicine Siriraj Hospital, Mahidol University, Bangkok, Thailand, 4 Department of Surgery, Faculty of Medicine Siriraj Hospital, Mahidol University, Bangkok, Thailand, 5 Institute of Cancer Sciences, University of Glasgow, Wolfson Wohl Cancer Research Centre, Garscube Estate, Glasgow, United Kingdom

☯ These authors contributed equally to this work.
* chanitra.thu@mahidol.ac.th (CT); Joanne.Edwards@glasgow.ac.uk (JE)

## Abstract

Colorectal cancer (CRC) is the third most common malignancy cause of cancer-related mortality worldwide. Epithelial-mesenchymal transition (EMT) promotes cancer metastasis and a tumour-based Glasgow EMT score was associated with adverse clinical features and poor prognosis. In this study, the impact of using the established five tumour-based EMT markers consisting of E-cadherin (E-cad), β-catenin (β-cat), Snail, Zeb-1, and Fascin in combination with the stromal periostin (PN) on the prediction of CRC patients' prognosis were invesigated. Formalin-fixed paraffin-embedded tissues of 202 CRC patients were studies the expressions of E-cad, β-cat, Snail, Zeb-1, Fascin, and PN by immunohistochemistry. Individually, cytoplasmic Fascin (Fc), cytoplasmic Snail (Sc), nuclear Snail (Sn), stromal Snail (Ss), and stromal PN (Ps) were significantly associated with reduced survival. A combination of Ps with Fc, Fs, and Sn was observed in 2 patterns including combined Fc, Fs, and Ps (FcFsPs) and Fc, Sn, and Ps (FcSnPs). These combinations enhanced the prognostic power compared to individual EMT markers and were independent prognostic markers. As the previously established scoring method required five markers and stringent criteria, its clinical use might be limited. Therefore, using these novel combined prognostic markers, either FcFsPs or FcSnPs, may be useful in predicting CRC patient outcomes.

## Introduction

Globally, colorectal cancer (CRC) is the third most commonly diagnosed malignancy and the fifth cause of cancer-related mortality, followed by lung cancer, liver cancer, gastric cancer,

**Data Availability Statement:** All the other data generated or analyzed during this study are included in this published article and its supplementary information files, and deposited in Mendeley Data repository DOI: 10.17632/wkhpvjghrw.1. The code can be shared upon request to the first or corresponding authors.

**Funding:** Siriraj Research Fund, Grant number (IO) R016533013, Faculty of Medicine Siriraj Hospital, Mahidol Universityd "The funders had no role in study design, data collection, and analysis, decision to publish, or preparation of the manuscript." in the Funding section.

**Competing interests:** The authors have declared that no competing interests exist.

and breast cancer [1]. Although the prognosis of early-stage CRC patients has improved greatly due to the advanced detection and treatment, advanced CRC patients still exhibit a poor prognosis with a 5-year survival rate of less than 20%, due to the presence or development of metastasis [2]. Approximately, 20% of CRC patients have metastasis at diagnosis and more than half of patients will develop metastases and relapse after initial treatment [2, 3].

The epithelial-mesenchymal transition (EMT) is a process of phenotypic transformation where epithelial cells lose cell-cell junction and polarity to become more stem-like or mesenchymal cells [4]. In cancer, the EMT process is believed to play a crucial role in treatment resistance and metastasis [4]. There are several proteins involved in EMT such as E-cadherin (E-cad), β-catenin (β-cat), Fascin, Snail, Zeb-1, and periostin (PN) [5, 6]. Each of these markers has previously been reported to have prognostic value in CRC [6]. In normal cells, membranous E-cad forms a complex with β-cat, preventing its nuclear translocation and the initiation of EMT [7]. In the metastatic setting, downregulation of E-cad allows β-cat to translocate into the nucleus and promotes upregulation of EMT-related gene expressions such as Snail, and Fascin [8, 9]. PN is a matricellular protein found in the periosteum and involved in bone extracellular matrix remodeling [10]. Aberrant expression of PN has been observed in various cancers and its expression correlates with shortened patient survival time [10]. Studies have reported that PN derived from cancer-associated fibroblasts can promote cancer cell metastasis [11, 12].

The prognostic value of individual EMT markers has long been recognized in CRC patients [9, 13–16]. Previously, we have reported the tumour-based EMT score consisting of a combination of E-cad, β-cat, Snail, Zeb-1, and Fascin expression and the association of this score with the adverse clinical features and poor prognosis in Glasgow CRC patients [6]. Furthermore, the expression of PN in the stroma of CRC tissues was correlated with the presence of metastasis and reduced survival time in CRC patients [17–20]. The high expression of PN in stromal cells in CRC was reported to mediate both tumorigenesis and tumor progression via several signaling pathways [17, 21, 22].

Though these markers can independently stratify the prognosis of CRC patients, it is still unclear whether PN in combination with the previously reported five tumour-based EMT markers can enhance the prognostic power in CRC patients or enable EMT markers to be reduced in number for clinical utilization. In the current study, we investigated the impact of using the established five tumour-based EMT markers, stromal PN, and the combination of stromal PN, and EMT markers for better CRC patients' prognosis. Stromal PN in combination with cytoplasmic, and stromal Fascin, and nuclear Snail was associated with patients' survival. These findings could be translated to an immunohistochemistry staining panel including only Fascin and Snail in the tumour cells and stromal PN to aid in clinician decision-making.

## Materials and methods

### Patient samples

Two hundred and two CRC specimens were identified retrospectively from Siriraj Hospital diagnosed which were the surgical samples from patients who signed the consent for surgical procedures between 2009 to 2015. Matched clinical data and formalin-fixed paraffin-embedded (FFPE) tissues were obtained under the collection protocol approved by Siriraj Hospital Institutional Review Board, COA no. Si 544/2015 and Si 628/2021. These surgical samples were then kept at Department of Pathology, Faculty of Medicine Siriraj Hospital as the hospital regulation. To use these samples, we asked for the permission from the hospital director through the content included in the process of ethical approval by the institutional review

board. Patients who received neoadjuvant chemotherapy and/or died within 30 days of surgery were excluded from the study.

## Immunohistochemistry of E-cad, β-cat, Fascin, Snail, Zeb-1, and PN

Antibody specificity and the staining protocols for the five EMT markers have been previously described [6, 17]. Briefly, tissue sections were deparaffinized in Histoclear and rehydrated through graded alcohols. Antigen retrieval was performed as follows; E-cad, Fascin, Snail, and Zeb-1 in citrate buffer pH 6.0, and PN in tris-EDTA buffer pH 8.0 under pressure for 5 min, whereas β-cat was retrieved in tris-EDTA buffer pH 8.0 in 96˚C water bath for 50 min. Endogenous peroxidase activity was quenched using 0.3% hydrogen peroxide ($H_2O_2$) for 30 min for β-cat, 3% $H_2O_2$ for 20 min for E-cad, Snail and Zeb-1, and 3% $H_2O_2$ for 10 min for PN. 10% casein (Vector Laboratories) was used to block for non-specific binding for all markers (30 min for Snail, Zeb-1, Fascin and PN, and 2 h for E-cadherin) except β-cat for which 1% bovine serum albumin was used (30 min). E-cad (1:500; BD Biosciences, 610182), Zeb-1 (1:800; Sigma-Aldrich, HPA027524), and PN (15 μg/ml; R&D system, AF3548) were incubated at overnight 4˚C. β-catenin (1:1000; BD Biosciences, 610154), Fascin (1:100; Atlas Antibodies, HPA005723), and Snail (1:50; Abcam, ab53519) were incubated at room temperature for 2 h. After washing, sections were incubated in goat anti-mouse or rabbit ImmPRESS (Vector Laboratories) for 30 min for E-cad, Zeb-1, and Fascin, and 20 min for β-cat. For Snail and PN, sections were incubated in horse anti-goat ImmPRESS (Vector Laboratories) for 30 min. The signal was visualized by 3,3'-diaminobenzidine (DAB; Vector Laboratories) for 3 min for E-cad and β-cat, and 5 min for other markers. All sections were counterstained in hematoxylin before being dehydrated through graded alcohols and Histoclear. The sections were mounted in DPX mountant.

## Immunohistochemistry scoring

The stained sections were scanned using Hamamatsu NanoZoomer *(*Welwyn Garden City, Hertfordshire, UK*)* at 20x magnification and visualized on NDP viewer *(*NanoZoomer Digital Pathology software, Hamamatsu Photonics K.K.*)*. Staining for the five EMT markers was assessed by a single examiner (NJ), blinded to clinicopathological data, using a weighted histoscore [23], calculated as follows: *(*% tumour with no staining x 0*) + (*% tumour with weak staining x1*) + (*% tumour with moderate staining x 2*) + (*% tumour with strong staining x3*)*, giving a range of scores between 0 and 300 for each marker at the membrane, cytoplasm, nucleus, and stroma compartments. Ten percent of sections were also co-scored by a coinvestigator (JE) and the interclass correlation coefficient was calculated to be > 0.7 for all markers. For PN staining, the scoring values were evaluated by the percentage of positive stromal cells *(*P*)* and the intensity of the staining signal *(*I*)*. For P, 0–25%, 26–50%, 51–75%, and 76–100% were classified as grades 0, 1, 2, and 3. For I, unstained, slightly stained, intermediately stained, and strongly stained were classified as 0, 1, 2, and 3. The expression scoring was calculated by P x I which covered the total score of 0–9. Two investigators performed this scoring double-blind to the clinicopathological data and each other (CT and PT).

## Statistical analysis

The expression of protein at each cellular compartment was divided into low and high using a publicly available R package in R-Studio to calculate the cut-off value for each protein. The correlation between protein expression and clinicopathological factors was accessed by Fisher's exact or Pearson's $\chi^2$ test. The Kaplan-Meier log-rank test was used for the survival function. The univariate Cox regression was used to calculate the hazard ratio *(*HR*)* and 95%

confidence interval *(95% CI)*. The step-wise backward conditional Cox regression was used to identify the independent prognostic marker. All statistical analyses were performed in SPSS version 23. The P-value < 0.05 was considered statistically significant.

## Results

### Immunohistochemical staining of E-cad, β-cat, Fascin, Snail, Zeb-1 and stromal PN in CRC tissues

E-cad, B-cat and Fascin expression was mainly observed in the membrane but some expression was also seen in the cytoplasm, nucleus and stroma *(Fig 1A–1C)*. For Snail and Zeb-1, Expression was commonly observed in the cytoplasm, although nuclear and stromal expression was also detected *(Fig 1D and 1E)*. The expression of PN was detected only in the stroma compartment *(Fig 1F)*.

### Demographic data and clinicopathological factor correlation

A total of 202 FFPE full sections from this Thai CRC patient cohort were included in this study. The number of sections stained for each marker was 148 for E-cad, Snail, and Zeb-1, 152 for β-cat, 192 for Fascin, and 202 for PN.

The demographic data for these sections were summarized (Table 1). The patient's ages ranged from 32 to 94 years with a median age of 64 years (Table 1). Around 54% of patients were younger than 65 years at the time of diagnosis with 56.4% of patients being male. The percentage of patients at stages I, II, III, and IV were 16.8%, 25.9%, 30.8%, and 26.5%, respectively. Local metastasis was present in 25.3% and distant metastasis was observed in 38.4% and 34.9% though lymphovascular and perineural routes. At the last follow up, 128 patients were alive, 57 patients were died by cancer-related death, and 17 patients were died by non-cancer related death. The median overall survival time was 49.9 months (range 1.5–104 months).

The correlation of E-cad, β-cat, Fascin, Snail, Zeb-1, and PN and patient survival time. The prognostic values of E-cad, β-cat, Fascin, Snail, Zeb-1, and PN expressions in different tissue compartments and cellular location were investigated (Table 2). Expression of Zeb-1 was not associated with prognosis in any tissue compartments (Table 2). The expression of tumour cytoplasmic Fascin (p = 0.019) (Fig 2A), tumour cytoplasmic Snail (p = 0.022) (Fig 2B), tumour nuclear Snail (p = 0.008) (Fig 2C), stromal Snail (p = 0.021) (Fig 2D), and stromal PN (p < 0.001) (Fig 2E) were associated with poor prognosis of the patients. By univariate analysis, tumour cytoplasmic Fascin score was significant predictive parameter of poor prognosis with HR 2.126 (95% CI 1.113–4.062, p = 0.022) (Table 2). Moreover, tumour cytoplasmic Snail (HR 1.994, 95% CI 1.091–3.646, p = 0.025), tumour nuclear Snail (HR 2.427, 95% CI 1.227–4.798, p = 0.011), stromal Snail (HR 2.136, 95% CI 1.102–4.140, p = 0.025), and stromal PN (HR 5.278, 95% CI 3.121–8.926, p < 0.001) were also associated with reduced survival time of the patients (Table 2).

Although not statistically significant, the expression of tumour membrane E-cad (HR 0.267, 95% CI 0.065–1.102, *p* = 0.050) (S1A Fig), and stromal Fascin (HR 0.584, 95% CI 0.337–1.011, *p* = 0.052) (S1B Fig), trended towards being good prognostic markers while tumour membrane β-cat trended towards being a poor prognostic marker (HR 1.829, 95% CI 0.986–3.391, *p* = 0.052) (S1C Fig). The relationship between each marker and clinicopathological factors is shown in S1–S6 Tables. Expression of membrane E-cad was associated with N stage (*p* = 0.002), cancer stage (*p* = 0.024), differentiation (*p* = 0.016), local metastasis (*p* = 0.041), lymphovascular invasion (*p* = 0.001), and perineural invasion (*p* = 0.010), and expression of stromal PN (*p* = 0.002), whereas nuclear E-cad was associated with M stage (*p* = 0.021), local metastasis (*p* = 0.007), and perineural invasion (*p* = 0.006) (S1 Table). For β-cat, the membrane

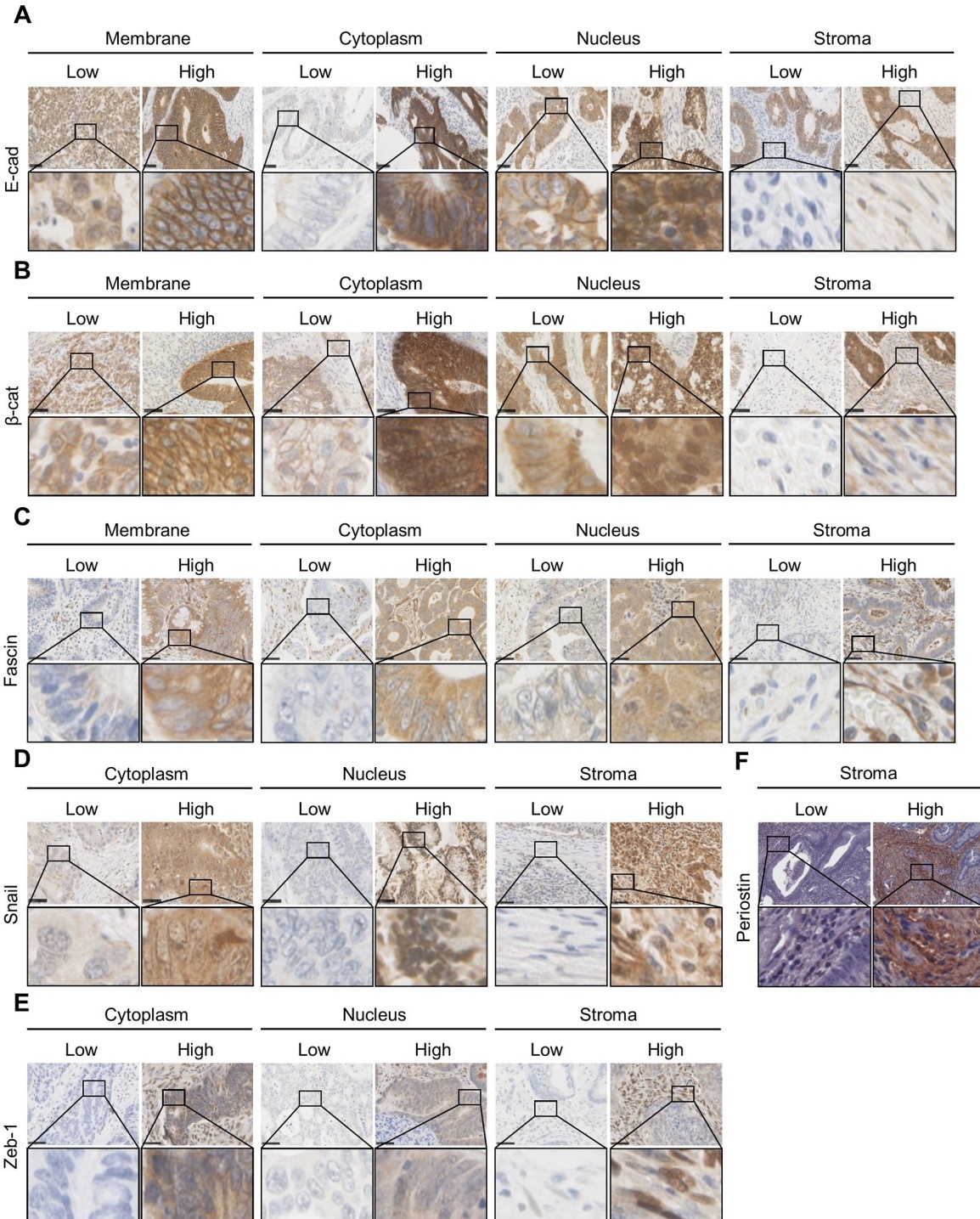

**Fig 1. Representative immunohistochemical staining of low and high expressions of tumour-based.** (**A**) E-cad, (**B**) β-cat, (**C**) Fas, (**D**) Snail, (**E**) Zeb-1, and (**F**) stromal PN in CRC tissues. The scale bar represents 100 μm. Original magnification of 200X.

expression was associated with M stage ($p$ = 0.013), cancer stage ($p$ = 0.015), and local metasta-sis ($p$ = 0.006); while the cytoplasmic expression was associated with cancer stage ($p$ = 0.014), and differentiation ($p$ = 0.014); nuclear β-cat expression was associated with M stage

**Table 1. Demographic data of CRC patients in this study.**

| Characteristics | n (%) |
|---|---|
| **Age (year) (n = 202)** | |
| <65 | 110 (54.5) |
| ≥65 | 92 (45.5) |
| **Sex (n = 202)** | |
| Female | 88 (43.6) |
| Male | 114 (56.4) |
| **Location (n = 202)** | |
| Colon | 108 (53.5) |
| Rectum | 94 (46.5) |
| **T stage (n = 185)** | |
| 1 | 9 (4.9) |
| 2 | 30 (16.2) |
| 3 | 113 (61.1) |
| 4 | 33 (17.8) |
| **N stage (n = 185)** | |
| 0 | 83 (44.9) |
| 1 | 56 (30.3) |
| 2 | 46 (22.9) |
| **M stage (n = 185)** | |
| 0 | 136 (73.5) |
| 1 | 49 (26.5) |
| **Stage (n = 185)** | |
| I | 31 (16.8) |
| II | 48 (25.9) |
| III | 57 (30.8) |
| IV | 49 (26.5) |
| **Differentiation (n = 190)** | |
| Well differentiation | 11 (5.8) |
| Moderately differentiation | 176 (92.6) |
| Poorly differentiation | 3 (1.6) |
| **Local metastasis (n = 190)** | |
| Absence | 142 (74.7) |
| Presence | 48 (25.3) |
| **Lymphovascular invasion (n = 198)** | |
| Absence | 122 (61.6) |
| Presence | 76 (38.4) |
| **Perineural invasion (n = 195)** | |
| Absence | 127 (65.1) |
| Presence | 68 (34.9) |

($p = 0.009$), cancer stage ($p = 0.024$), local metastasis ($p = 0.015$), and stromal PN ($p = 0.012$) (S2 Table). For Snail, no clinicopathological correlation was observed (S3 Table). For Fascin, membrane expression was associated with T stage ($p = 0.008$), N stage ($p = 0.025$), cancer stage ($p = 0.011$), and differentiation ($p = 0.003$); nuclear fascin was associated with N stage ($p = 0.023$); stromal Fascin expression with N stage ($p = 0.001$), and cancer stage ($p = 0.003$) (S4 Table). For Zeb-1, only stromal expression was correlated with N stage ($p = 0.025$), and cancer stage ($p = 0.009$) (S5 Table). Lastly, stromal PN was associated with T stage ($p = 0.002$),

**Table 2. Prognostic value of tumour-based E-cad, β-cat, Fascin, Snail, Zeb-1 and stromal PN in CRC samples.**

| Markers | n (%) | Events (cancer death) | Univariate HR (95% CI) | P-value | 5-year OS (%) |
|---|---|---|---|---|---|
| **E-cad** | | | | | |
| membrane low | 130 (87.8) | 45 | | | 63 |
| high | 18 (12.2) | 2 | 0.267 (0.065–1.102) | 0.068 | 94 |
| cytoplasm low | 131 (88.5) | 40 | | | 68 |
| high | 17 (11.5) | 7 | 1.611 (0.720–3.606) | 0.246 | 57 |
| nucleus low | 126 (85.1) | 43 | | | 64 |
| high | 22 (14.9) | 4 | 0.463 (0.166–1.291) | 0.141 | 81 |
| stroma low | 107 (72.3) | 37 | | | 64 |
| high | 41 (27.7) | 10 | 0.718 (0.357–1.444) | 0.353 | 74 |
| **β-cat** | | | | | |
| membrane low | 117 (77.0) | 31 | | | 71 |
| high | 35 (23.0) | 15 | 1.829 (0.986–3.391) | 0.056 | 57 |
| cytoplasm low | 76 (50.0) | 20 | | | 71 |
| high | 76 (50.0) | 26 | 1.332 (0.744–2.387) | 0.335 | 65 |
| nucleus low | 39 (25.7) | 8 | | | 78 |
| high | 113 (74.3) | 38 | 1.750 (0.816–3.753) | 0.150 | 65 |
| stroma low | 38 (25.0) | 9 | | | 73 |
| high | 114 (75.0) | 37 | 1.523 (0.734–3.158) | 0.259 | 66 |
| **Fascin** | | | | | |
| membrane low | 164 (85.4) | 46 | | | 71 |
| high | 28 (14.6) | 5 | 0.561 (0.223–1.411) | 0.219 | 82 |
| cytoplasm low | 72 (37.5) | 12 | | | 82 |
| high | 120 (62.5) | 39 | 2.126 (1.113–4.062) | **0.022** | 67 |
| nucleus low | 118 (61.5) | 34 | | | 69 |
| high | 74 (38.5) | 17 | 0.771 (0.431–1.380) | 0.381 | 77 |
| stroma low | 73 (38.0) | 25 | | | 63 |
| high | 119 (62.0) | 26 | 0.584 (0.337–1.011) | 0.055 | 78 |
| **Snail** | | | | | |
| cytoplasm low | 76 (51.4) | 17 | | | 77 |
| high | 72 (48.6) | 28 | 1.994 (1.091–3.646) | **0.025** | 59 |
| nucleus low | 127 (85.8) | 34 | | | 72 |
| high | 21 (14.2) | 11 | 2.427 (1.227–4.798) | **0.011** | 45 |
| stroma low | 61 (41.2) | 12 | | | 79 |
| high | 87 (58.8) | 33 | 2.136 (1.102–4.140) | **0.025** | 61 |
| **Zeb-1** | | | | | |
| cytoplasm low | 120 (81.1) | 38 | | | 67 |
| high | 28 (18.9) | 6 | 0.741 (0.313–1.757) | 0.497 | 78 |
| nucleus low | 84 (56.8) | 27 | | | 67 |
| high | 64 (43.2) | 17 | 0.839 (0.457–1.539) | 0.571 | 71 |
| stroma low | 108 (73.0) | 29 | | | 72 |
| high | 40 (27.0) | 15 | 1.477 (0.791–2.756) | 0.221 | 60 |
| **PN** | | | | | |
| stroma low | 151 (74.8) | 26 | | | 82 |
| high | 51 (25.2) | 31 | 5.278 (3.121–8.926) | **<0.001** | 37 |

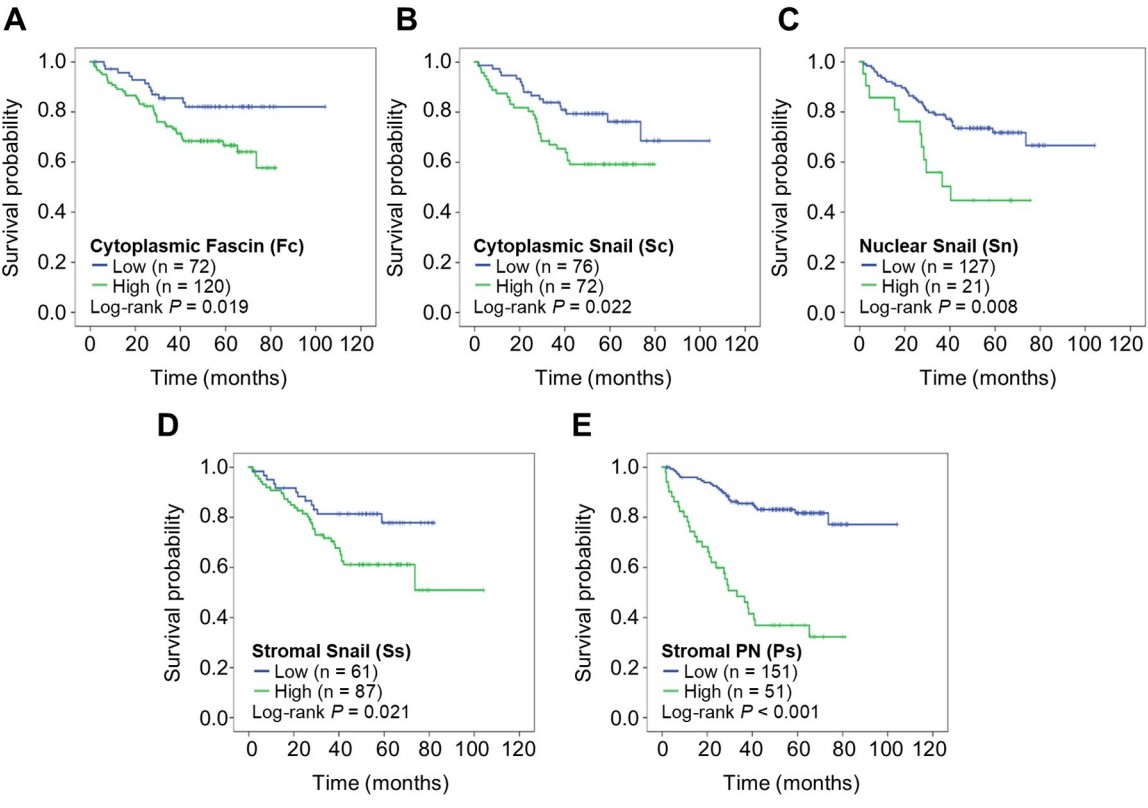

**Fig 2. KM-analysis.** (**A**) cytoplasmic Fas, (**B**) cytoplasmic Snail, (**C**) nuclear Snail, (**D**) stromal Snail, (**E**) stromal PN.

N stage ($P = 0.001$), M stage ($p < 0.001$), local metastasis ($p < 0.001$), lymphovascular invasion ($p < 0.001$), and perineural invasion ($p = 0.006$) (S6 Table).

## The combination of EMT markers and PN improves prognostic strength in CRC patients

The expression levels of cytoplasmic Fascin (Fc), cytoplasmic Snail (Sc), nuclear Snail (Sn), stromal Snail (Sn), membrane E-cad (Em), membrane β-cat (Bm), and cytoplasmic Fascin (Fc) were combined with stromal PN (Ps) to investigate the prognostic strength (S2A–S2F Fig). Patients with high Fc/high Ps ($p < 0.001$), high Sc/high Ps ($p < 0.001$), high Sn/high Ps ($p < 0.001$), and high Ss/high Ps ($p < 0.001$) had a shorter survival time with statistical significance (S2A–S2D Fig). However, low Em ($p = 0.003$) or low Fs ($p < 0.001$) with high Ps had a short survival time (S2E–S2F Fig). These combination patterns resulted in an improved prognostic value compared to individual markers, however, they could not statistically separate patients into good and poor prognosis groups.

Importantly, combining Ps with Fc, Fs, and Sn as FcFsPs and FcSnPs could efficiently stratify the prognosis of patients into good and poor prognostic groups (Fig 3A and 3B). *In the FcFsPs model*, patients could be grouped into; 1) "*good prognosis*" characterized by either low expression of all markers (low Fs/low Fc/low Ps) or high Fs with either low expression of both Fc and Ps (high Fs/low Fc/low Ps) or only one of them high (high Fs/high Fc/low Ps or high Fs/low Fc/high Ps); 2) "*intermediate 1*" characterized by low Fs with either high Fc or high Ps (low Fs/low Fc/high Ps or low Fs/high Fc/low Ps) (HR 3.368, 95% CI 1.488–7.626, $P = 0.004$); 3) "*intermediate 2*" characterized by high expression of all three markers (high Fs/high Fc/high

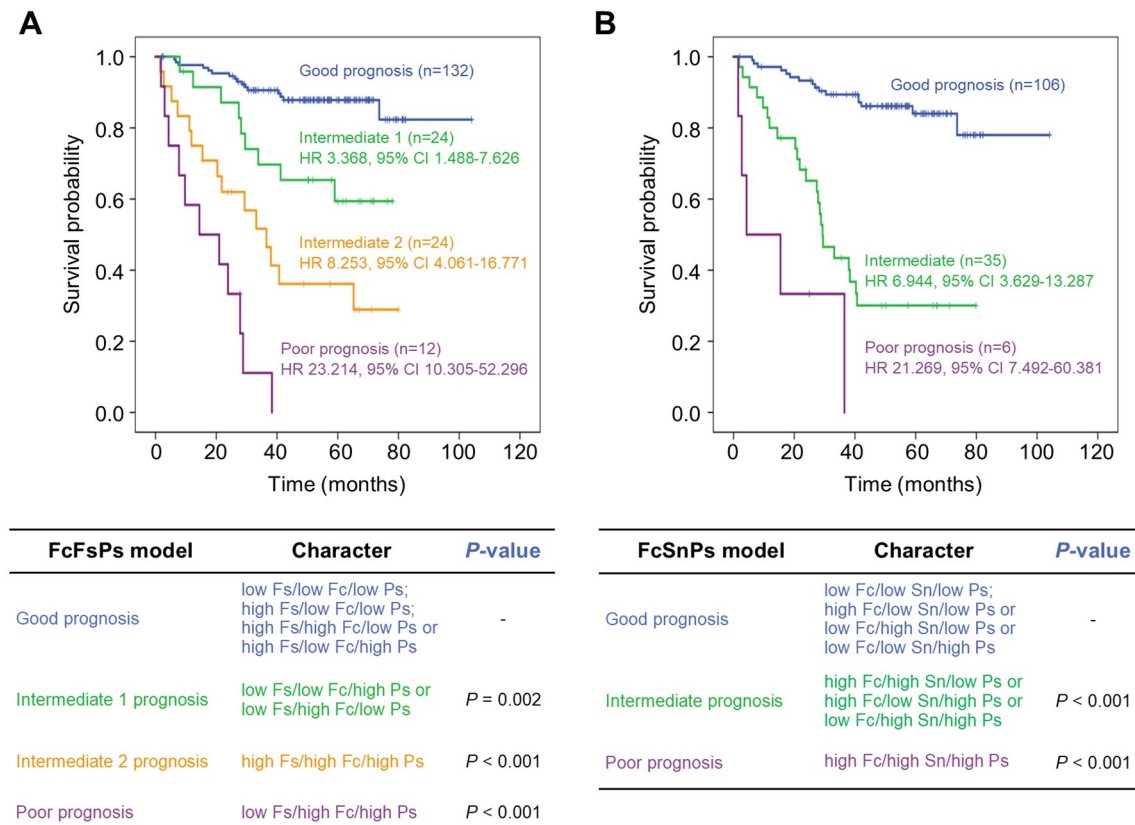

| FcFsPs model | Character | P-value |
|---|---|---|
| Good prognosis | low Fs/low Fc/low Ps; high Fs/low Fc/low Ps; high Fs/high Fc/low Ps or high Fs/low Fc/high Ps | - |
| Intermediate 1 prognosis | low Fs/low Fc/high Ps or low Fs/high Fc/low Ps | P = 0.002 |
| Intermediate 2 prognosis | high Fs/high Fc/high Ps | P < 0.001 |
| Poor prognosis | low Fs/high Fc/high Ps | P < 0.001 |

| FcSnPs model | Character | P-value |
|---|---|---|
| Good prognosis | low Fc/low Sn/low Ps; high Fc/low Sn/low Ps or low Fc/high Sn/low Ps or low Fc/low Sn/high Ps | - |
| Intermediate prognosis | high Fc/high Sn/low Ps or high Fc/low Sn/high Ps or low Fc/high Sn/high Ps | P < 0.001 |
| Poor prognosis | high Fc/high Sn/high Ps | P < 0.001 |

**Fig 3. KM-analysis of FcFsPs and FcSnPs models.**

Ps) (HR 8.253, 95% CI 4.061–16.771, $P < 0.001$); and 4) "*poor prognosis*" characterized by low expression of Fs with high expression of both Fc and Ps (low Fs/high Fc/high Ps) (HR 23.214, 95% CI 10.305–52.296, $P < 0.001$) (Fig 3A).

For the ***FcSnPs model***, patients were stratified into; 1) "*good prognosis*" which is either a low level of all markers (low Fc/low Sn/low Ps) or a high level of only one marker (high Fc/low Sn/low Ps or low Fc/high Sn/low Ps or low Fc/low Sn/high Ps); 2) "*intermediate prognosis*" characterized by high expression of two markers (high Fc/high Sn/low Ps or high Fc/low Sn/high Ps or low Fc/high Sn/high Ps) (HR 6.944, 95% CI 3.629–13.287, $p < 0.001$,); and 3) "*poor prognosis*" characterized by high expression of all markers (high Fc/high Sn/high Ps) (HR 21.269, 95% CI 7.492–60.381, $p < 0.001$) (Fig 3B). Interestingly, both FcFsPs (HR 1.753, 95% CI 1.256–2.448, $p = 0.001$) and FcSnPs (HR 2.178 95% CI 1.097–4.322, $p = 0.026$) models demonstrated independent prognostic value for CRC patients (Table 3).

Lastly, the FcFsPs and FcSnPs models were assessed for clinicopathological association (Table 4). Stratifying CRC patients based on the FcFsPs model was associated with high N/M and cancer stage, and presence of local metastasis ($p < 0.001$), lymphovascular invasion ($p = 0.006$), and perineural invasion ($p = 0.007$). For FcFsPs, the clinical parameters showing significant association were T stage ($p = 0.035$), N stage ($p = 0.008$), perineural invasion ($p = 0.011$), and M stage, cancer stage, local metastasis, lymphovascular invasion ($p < 0.001$).

## Discussion

The EMT process is considered one of the underlying mechanisms of CRC metastasis [24]. During EMT, not only do the cancer cells undergo genotypic and phenotypic alteration but

**Table 3. Univariate and multivariate Cox regression analyses of FcFsPs model and FcSnPs models in CRC samples.**

| Covariates | Forced-entry | | Backward conditional | | | |
| --- | --- | --- | --- | --- | --- | --- |
| | HR (95% CI) | *P*-value | FcFsPs model | | FcSnPs model | |
| | | | HR (95% CI) | *P*-value | HR (95% CI) | *P*-value |
| **Age** (<65/≥65) | 0.748 (0.439–1.275) | 0.285 | | | | |
| **Sex** (male/female) | 1.214 (0.715–2.061) | 0.474 | | | | |
| **Location** (colon/rectum) | 0.988 (0.587–1.662) | 0.963 | | | | |
| **T score** (0/1/2/3) | 3.209 (2.042–5.045) | **<0.001** | 2.895 (1.450–5.777) | **0.003** | 2.221 (1.116–4.418) | 0.076 |
| **N score** (0/1/2) | 2.682 (1.900–3.787) | **<0.001** | | | | |
| **M score** (0/1) | 13.829 (7.264–26.328) | **<0.001** | 8.132 (3.349–19.747) | **<0.001** | 8.484 (2.933–24.542) | **<0.001** |
| **Stage** (I/II/III/IV) | 6.368 (3.718–10.908) | **<0.001** | | | | |
| **Differentiation** (well/moderately/poorly) | 4.143 (1.829–9.380) | **0.001** | 39.687 (5.186–303.720) | **<0.001** | 25.890 (3.367–199.056) | **0.002** |
| **Local metastasis** (absence/presence) | 12.688 (6.605–24.374) | **<0.001** | | | | |
| **Lymphovascular metastasis** (absence/presence) | 3.457 (2.010–5.946) | **<0.001** | | | 2.514 (1.085–5.823) | **0.031** |
| **Perineural metastasis** (absence/presence) | 3.460 (2.034–5.887) | **<0.001** | | | | |
| **FcFsPs** | 2.837 (2.210–3.642) | **<0.001** | 1.753 (1.256–2.448) | **0.001** | | |
| **FcSnPs** | 5.155 (3.326–7.992) | **<0.001** | | | 2.178 (1.097–4.322) | **0.026** |

also the tumor microenvironment (TME) surrounding the cancer cell changes. The interplay between cancer cells and TME during EMT allows the cancer cells to metastasize [25, 26]. Metastatic CRC has a poor prognosis and is inherently more resistant to chemotherapy treatment [2, 27]. Therefore, predicting the metastatic event in CRC patients based on the changes in EMT and TME could help identify CRC patients who may require more aggressive treatment. In the present study, high expression of Fc, Sc, Sn, Ss, and Ps was individually associated with a reduced survival time in CRC patients. Furthermore, the combination of Fc, Fs, and Fs (FcFsPs) and Fc, Sn, and Ps (FcSnPs) could efficiently stratify prognosis for CRC patients. The combined markers demonstrated a higher prognostic strength when compared to using an individual marker and may provide a novel prognosis tool for CRC patients.

The expression of transcriptional factor Snail in cancer cells has long been recognized as a marker of poor prognosis for multiple malignancies including CRC [8, 16, 28]. Snail is known as a negative regulator of E-cad at the transcriptomic level, an initiation event for cancer metastasis [29, 30]. Studies have reported that the presence of Snail in ovarian cancer contributes to immunosuppression by up-regulating CXCL1 and CXCL2 expression which promotes recruitment of myeloid-derived suppressor cells (MDSCs) [31, 32]. Furthermore, expression of Snail in esophageal squamous cell carcinoma also promotes PD-L1 expression that induces T cell apoptosis [31, 32]. The pro-tumorigenic function and prognostic value of Snail expression was also found in stromal cells [33–35]. In CRC, Snail-expressing cancer-associated fibroblasts (CAFs) showed different cytokines secretion profiles including CCL1, CCL7, and CXCL1 when compared to normal fibroblasts affecting CRC cell migration [36]. Furthermore, co-injection of Snail-knockout fibroblast and CRC cells demonstrated low proliferative index by ki67 staining, high expression of membrane E-cad in the CRC cells, and reduced local

**Table 4. Chi-square analysis of FcFsPs model and FcSnPs models in CRC samples, Int: intermediate.**

| Clinical parameters | | FcFsPs | | | | | FcSnPs | | | |
|---|---|---|---|---|---|---|---|---|---|---|
| | | Good (n = 132) | Int 1 (n = 24) | Int 2 (n = 24) | Poor (n = 12) | $\chi^2$ | Good (n = 106) | Int (n = 35) | Poor (n = 6) | $\chi^2$ |
| **Age** | <65 | 71(53.8) | 14(58.3) | 12(50.0) | 7(58.3) | 0.935 | 60(56.6) | 19(54.3) | 5(83.3) | 0.405 |
| | ≥65 | 61(46.2) | 10(41.7) | 12(50.0) | 5(41.7) | | 46(43.4) | 16(45.7) | 1(16.7) | |
| **Sex** | Female | 55(41.7) | 17(70.8) | 5(20.8) | 8(66.7) | **0.002** | 49(46.2) | 1748.6) | 0(0.0) | 0.076 |
| | Male | 77(58.3) | 7(29.2) | 19(79.2) | 4(33.3) | | 57(53.8) | 18(51.4) | 6(100.0) | |
| **Location** | Colon | 71(53.8) | 12(50.0) | 12(50.0) | 8(66.7) | 0.783 | 58(54.7) | 18(51.4) | 3(50.0) | 0.928 |
| | Rectum | 61(46.2) | 12(50.0) | 12(50.0) | 4(33.3) | | 48(45.3) | 17(48.6) | 3(50.0) | |
| **T stage** | 1 | 8(6.1) | 1(4.2) | 0(0.0) | 0(0.0) | 0.133 | 9(8.5) | 0(0.0) | 0(0.0) | **0.035** |
| | 2 | 27(20.5) | 1(4.2) | 2(8.3) | 1(8.3) | | 20(18.9) | 3(8.6) | 0(0.0) | |
| | 3 | 81(61.4) | 15(62.5) | 17(70.8) | 7(58.3) | | 65(61.3) | 24(68.6) | 3(50.0) | |
| | 4 | 16(12.1) | 7(29.2) | 5(20.8) | 4(33.3) | | 12(11.3) | 8(22.9) | 3(50.0) | |
| **N stage** | 0 | 72(54.5) | 3(13.0) | 6(25.0) | 0(0.0) | <**0.001** | 53(50.0) | 7(20.0) | 1(16.7) | **0.008** |
| | 1 | 34(25.8) | 12(52.2) | 9(37.5) | 5(41.7) | | 31(29.2) | 12(34.3) | 3(50.0) | |
| | 2 | 26(19.7) | 8(34.8) | 9(37.5) | 7(58.3) | | 22(20.8) | 16(45.7) | 2(33.3) | |
| **M stage** | 0 | 111(85.4) | 16(69.6) | 8(40.0) | 1(11.1) | <**0.001** | 89(84.8) | 10(33.3) | 1(33.3) | <**0.001** |
| | 1 | 19(14.6) | 7(30.4) | 12(60.0) | 8(88.9) | | 16(15.2) | 20(66.7) | 2(66.7) | |
| **Stage** | I | 29(22.5) | 0(0.0) | 0(0.0) | 0(0.0) | <**0.001** | 22(21.4) | 0(0.0) | 0(0.0) | <**0.001** |
| | II | 42(32.6) | 3(13.6) | 2(12.5) | 0(0.0) | | 30(29.1) | 4(14.8) | 0(0.0) | |
| | III | 39(30.2) | 12(54.5) | 2(12.5) | 1(11.1) | | 3(34.0) | 3(11.1) | 0(0.0) | |
| | IV | 19(14.7) | 7(31.8) | 12(75.0) | 8(88.9) | | 1(15.5) | 20(74.1) | 2(100.0) | |
| **Differentiation** | Well | 10(7.9) | 0(0.0) | 0(0.0) | 0(0.0) | 0.133 | 9(8.9) | 0(0.0) | 0(0.0) | 0.155 |
| | Moderately | 115(91.3) | 21(100.0) | 23(95.8) | 11(91.7) | | 9(90.1) | 32(94.1) | 6(100.0) | |
| | Poorly | 1(0.8) | 0(0.0) | 1(4.2) | 1(8.3) | | 1(1.0) | 2(5.9) | 0(0.0) | |
| **Local metastasis** | Absence | 11(85.4) | 16(72.7) | 8(40.0) | 1(11.1) | <**0.001** | 89(85.6) | 1(32.3) | 1(33.3) | <**0.001** |
| | Presence | 19(14.6) | 6(27.3) | 12(60.0) | 8(88.9) | | 15(14.4) | 21(67.7) | 2(66.7) | |
| **Lymphovascular invasion** | Absence | 89(68.5) | 13(54.2) | 11(45.8) | 3(25.0) | **0.006** | 72(69.2) | 11(31.4) | 4(66.7) | <**0.001** |
| | Presence | 41(31.5) | 11(45.8) | 13(54.2) | 9(75.0) | | 32(30.8) | 24(68.6) | 2(33.3) | |
| **Perineural invasion** | Absence | 93(73.2) | 13(56.5) | 12(50.0) | 4(33.3) | **0.007** | 72(70.6) | 15(42.9) | 3(50.0) | **0.011** |
| | Presence | 34(26.8) | 10(43.5) | 12(50.0) | 8(66.7) | | 30(29.4) | 20(57.1) | 3(50.0) | |

metastatic lesion when compared to co-injection with Snail-wild type fibroblast [36]. Therefore, Snail expression has been identified in both cancer and stromal cells and both sources could function in concert to promote tumor progression. This evidence supports the findings in this study that high Sn predicts poor prognosis in CRC patients.

Fascin is an actin filament-bundling protein that is crucial for the maintenance of cell structure and secretory function of cells [37]. In cancer, re-arrangement of the cytoskeleton is a prerequisite event before metastasis [9, 37]. Studies in CRC tissues demonstrated that Fascin expression was low in normal colon mucosa, but overexpressed in the cytoplasm of CRC cells [38, 39]. Moreover, the overexpression of Fascin in CRC is associated with distant metastasis, reduced survival time, and disease recurrence [39–41]. These data support our findings that high Fascin in the cytoplasm of cancer cells is associated with poor prognosis of CRC patients.

In contrast to the expression in cancer cells, Fascin in the stroma of CRC was reported as a marker of good prognosis and that elevated expression inversely correlated with the stage of cancer and lymph node metastasis [42]. This finding was consistent with the good prognostic value of stromal Fascin (Fs) observed in the present study. In endometrial neoplasia development, the expression of stromal Fascin was reduced while its increased expression was

observed in endometrioid carcinoma; and the loss of stromal Fascin was also associated with a higher grade of endometrioid carcinoma [42]. However, in the ovarian cancer model, knockdown or pharmacologic inhibition of Fascin in stromal cells was shown to reduce cell migration and may contribute to delayed metastatic events [43]. Furthermore, a recent study using small molecule inhibitors specific to Fascin has shown that tumor-bearing mice treated with Fascin inhibitors and anti-program death-1 (PD-1) reduced tumor growth [44]. Fascin inhibitors impede dendritic cell migration out from the tumor bed and enhance their antigen-uptake ability subsequently increasing T cells activation that controls tumor growth [44]. Within tumor stroma, there are several types of cells and differential expression of Fascin in these cells may contribute to the different findings regarding the prognostic value of stromal Fascin and warrant further investigation.

PN or osteoblast-specific factor 2 (OSF-2) is a secreted protein mainly found in osteoblast and functions in cell-matrix interactions, and cell differentiation [10]. In CRC, PN was reported to be overexpressed in stromal cells and associated with aggressive clinical features and poor prognosis [18–20]. Mechanistically, PN was found to activate integrin $\alpha5\beta1$ or $\alpha6\beta4$ and subsequently stimulate Akt/PI3K resulting in enhanced chemotherapy resistance, cell migration, and metastatic properties [17, 21, 45]. Studies have demonstrated that PN secreted by stromal cells of various cancer can also modulate the expression of EMT-associated proteins such as $\beta$-cat, E-cad, and Zeb-1, supporting cancer metastasis [46–48]. All of these data strongly support the finding that high PN in the stromal cells of CRC tissues is indicative of poor prognosis. Taken all together, the current study shows that by the following combinations of protein markers is predictive of poor prognosis in CRC patients, 1) low stromal Fascin/high both cytoplasmic Fascin and stromal PN (low Fs/high FcPs) and 2) high cytoplasmic Fascin/nuclear Snail/stromal PN (high FcSnPs). However, the potential approach could involve incorporating the FcFsPs/FcSn/Ps into the existing molecular classification framework. By assessing the expression levels of these markers alongside genetic signatures, a more comprehensive understanding of CRC subtypes and their prognostic implications may be achieved. Furthermore, considering the evident interaction between stromal and cancer cells highlighted in our study, there's a strong rationale for expanding the current classification system to encompass aspects of the tumor microenvironment. This could involve developing new classification schemes that incorporate both genetic and pathological signatures, thus providing a more holistic view of CRC biology and potentially refining treatment strategies.

Although individual EMT markers have prognostic value, the combination of multiple EMT markers has been reported as a superior tool for CRC clinical outcome stratification [6, 49]. In a previous study, Roseweir et al. demonstrated that by combing membrane E-cad, nuclear $\beta$-cat, and cytoplasmic Fascin, Snail, and Zeb-1 CRC patients could be stratified into three categories (absence, low, and high) of prognosis [6]. However, based on this previous EMT scoring criteria, the number of samples categorized as absent or high was too low to perform statistical analysis. Additionally, as the scoring method required five markers and stringent criteria, its clinical use might be limited. The retrospective paraffin-embedded tissues used in this study mostly had degraded mRNA from a long storage. This limitation restricts our ability to directly measure mRNA level. Moreover, the quantification of FcFsPs/FcSnPs to be guide the clinical practice is our future study. In comparison, the present study proposes a novel prognostic marker combination that requires only two or three markers to stratify CRC patients which could be more practical in the clinic. Both FcFsPs and FcSnPs models are proposed as independent prognosis factors and demonstrate superior prognostic strength.

## Supporting information

**S1 Fig. KM-analysis of membrane E-cad, stromal Fas, and membrane β-cat.**
(TIF)

**S2 Fig. KM-analysis of PN in combination with EMT markers.**
(TIF)

**S1 Table. Association of tumour-based E-cad with clinicopathological factors of Thai CRC patients.**
(XLSX)

**S2 Table. Association of tumour-based β-cat with clinicopathological factors of Thai CRC patients.**
(XLSX)

**S3 Table. Association of tumour-based Snail with clinicopathological factors of Thai CRC patients.**
(XLSX)

**S4 Table. Association of tumour-based Fascin with clinicopathological factors of Thai CRC patients.**
(XLSX)

**S5 Table. Association of tumour-based Zeb-1 with clinicopathological factors of Thai CRC patients.**
(XLSX)

**S6 Table. Association of stromal PN with clinicopathological factors of Thai CRC patients.**
(XLSX)

## Acknowledgments

Special thanks to Surat Phumphuang for clinical data collection.

## Author Contributions

**Conceptualization:** Niphat Jirapongwattana, Suyanee Thongchot, Chanitra Thuwajit, Joanne Edwards.

**Data curation:** Niphat Jirapongwattana.

**Formal analysis:** Niphat Jirapongwattana, Suyanee Thongchot.

**Methodology:** Niphat Jirapongwattana, Suyanee Thongchot.

**Project administration:** Chanitra Thuwajit.

**Resources:** Ananya Pongpaibul, Atthaphorn Trakarnsanga.

**Software:** Niphat Jirapongwattana, Jean Quinn, Peti Thuwajit.

**Supervision:** Chanitra Thuwajit, Joanne Edwards.

**Validation:** Niphat Jirapongwattana, Jean Quinn, Peti Thuwajit.

**Visualization:** Jean Quinn, Peti Thuwajit.

**Writing – original draft:** Niphat Jirapongwattana, Suyanee Thongchot, Chanitra Thuwajit, Joanne Edwards.

**Writing – review & editing:** Niphat Jirapongwattana, Suyanee Thongchot, Chanitra Thuwajit, Joanne Edwards.

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
