## [Decision Letter · Decision Letter 0]

1 May 2024

PONE-D-23-41841The combined tumour-based Fascin/Snail and stromal periostin reveals the effective prognosis prediction in colorectal cancer patientsPLOS ONE

Dear Dr. Thuwajit,

Thank you for submitting your manuscript to PLOS ONE. After careful consideration, we feel that it has merit but does not fully meet PLOS ONE’s publication criteria as it currently stands. Therefore, we invite you to submit a revised version of the manuscript that addresses the points raised during the review process.

We look forward to receiving your revised manuscript.

Kind regards,

Peng Zhang, Ph.D.

Academic Editor

PLOS ONE

Journal Requirements:

"Siriraj Research Fund, Grant number (IO) R016533013, Faculty of Medicine Siriraj Hospital, Mahidol University"       

7. When completing the data availability statement of the submission form, you indicated that you will make your data available on acceptance. We strongly recommend all authors decide on a data sharing plan before acceptance, as the process can be lengthy and hold up publication timelines. Please note that, though access restrictions are acceptable now, your entire data will need to be made freely accessible if your manuscript is accepted for publication. This policy applies to all data except where public deposition would breach compliance with the protocol approved by your research ethics board. If you are unable to adhere to our open data policy, please kindly revise your statement to explain your reasoning and we will seek the editor's input on an exemption. Please be assured that, once you have provided your new statement, the assessment of your exemption will not hold up the peer review process.

Reviewers' comments:

Reviewer's Responses to Questions

**Comments to the Author**

1. Is the manuscript technically sound, and do the data support the conclusions?

Reviewer #1: Yes

Reviewer #2: Yes

2. Has the statistical analysis been performed appropriately and rigorously? 

Reviewer #1: Yes

Reviewer #2: Yes

3. Have the authors made all data underlying the findings in their manuscript fully available?

Reviewer #1: Yes

Reviewer #2: Yes

4. Is the manuscript presented in an intelligible fashion and written in standard English?

Reviewer #1: Yes

Reviewer #2: Yes

5. Review Comments to the Author

Reviewer #1: 1. Overall a good attempt.

2. However the quality of images is poor. Resend high resolution images for better understanding of the research findings.

3. Based on your research work, what are the recommendations of authors regarding modification in molecular classification of Colorectal cancer ?

Reviewer #2: This study is clearly structured and logical. Interesting data analyses are presented in the results section and are consistent with the research questions. However, it lacks an effective form of visual presentation. And the protein prognosis studied in the paper needs to be compared with the mRNA prognosis. It would be better if FcFsPs/FcSnPs could be quantified to guide the clinic. Overall, I think your study is of some importance to the field and has potential for publication!

6. PLOS authors have the option to publish the peer review history of their article (what does this mean?). If published, this will include your full peer review and any attached files.

Reviewer #1: No

Reviewer #2: **Yes: **Dunhui Yang

---

## [Author Response · Author response to Decision Letter 0]

14 May 2024

PONE-D-23-41841

The combined tumour-based Fascin/Snail and stromal periostin reveals the effective prognosis prediction in colorectal cancer patients

Please, see below how we have addressed item-by-item the reviewer’s criticisms. The “Revised Manuscript with Track Changes” file is included.

Yours Sincerely

Chanitra Thuwajit, Ph.D., M.D. (corresponding author, on behalf of all authors)

Journal Requirements:

Answer: Thank you very much for your valuable suggestion on this important point. We carefully checked and confirmed that we have adhered to the following PLOS ONE style templates.

Answer: In this manuscript, we used a publicly available R package to calculate the cut-off value for each protein. The sentences were added in the Section of Statistical analysis, yellow highlight. The code can be shared upon request to the first or corresponding authors mentioned in the Section of Data Availability Statement, yellow highlight. 

Answer: Thank you for this important point. The following paragraph is added in the ethical statement in the Materials and methods section, yellow highlight.

 These retrospective FFPE samples were the surgical samples from patients signing the consent forms for surgical procedures. These surgical samples were then kept at the Department of Pathology, Faculty of Medicine Siriaj Hospital. To use these, permission from the hospital director in the process of ethical approval by the institutional review board was needed.

Answer: We have removed the phrase referring to these data as suggested. Thank you for a suggestion.

"Siriraj Research Fund, Grant number (IO) R016533013, Faculty of Medicine Siriraj Hospital, Mahidol University" 

Answer: The sentence “The funders had no role in study design, data collection, and analysis, decision to publish, or preparation of the manuscript.” was added in the cover letter and section of Funding, yellow highlight. 

Answer: Thank you. The ethics statement appeared only in the Methods section under the topic “Patient samples” in the revised manuscript.

7. When completing the data availability statement of the submission form, you indicated that you will make your data available on acceptance. We strongly recommend all authors decide on a data sharing plan before acceptance, as the process can be lengthy and hold up publication timelines. Please note that, though access restrictions are acceptable now, your entire data will need to be made freely accessible if your manuscript is accepted for publication. This policy applies to all data except where public deposition would breach compliance with the protocol approved by your research ethics board. If you are unable to adhere to our open data policy, please kindly revise your statement to explain your reasoning and we will seek the editor's input on an exemption. Please be assured that, once you have provided your new statement, the assessment of your exemption will not hold up the peer review process.

Answer: We appreciate the reviewer’s insightful suggestion. We do agree and decide on a data-sharing plan before acceptance.

Answer: Thank you very much. This concern is understandable. We have already checked and removed ref. no. 49 (old) that have been retracted from our manuscript. The ref. no. 50 has changed to be ref. no. 49 (new).

Reviewers' comments:

Reviewer's Responses to Questions

Comments to the Author

1. Is the manuscript technically sound, and do the data support the conclusions?

Reviewer #1: Yes

Reviewer #2: Yes

2. Has the statistical analysis been performed appropriately and rigorously?

Reviewer #1: Yes

Reviewer #2: Yes

3. Have the authors made all data underlying the findings in their manuscript fully available?

Reviewer #1: Yes

Reviewer #2: Yes

4. Is the manuscript presented in an intelligible fashion and written in standard English?

Reviewer #1: Yes

Reviewer #2: Yes

5. Review Comments to the Author

Reviewer #1: 

1. Overall a good attempt.

Answer: Thank you very much. We are grateful for this positive assessment of our work.

2. However the quality of images is poor. Resend high resolution images for better understanding of the research findings.

Answer: We appreciate the reviewer’s insightful suggestion. New images with high resolution have already been submitted.

3. Based on your research work, what are the recommendations of authors regarding modification in molecular classification of Colorectal cancer?

Answer: Thank you for this valuable comment. To address this point, the following paragraph is added in Section of Discussion, yellow highlight.

 The potential approach could involve incorporating the FcFsPs/FcSn/Ps into the existing molecular classification framework. By assessing the expression levels of these markers alongside genetic signatures, a more comprehensive understanding of CRC subtypes and their prognostic implications may be achieved. Furthermore, considering the evident interaction between stromal and cancer cells highlighted in our study, there's a strong rationale for expanding the current classification system to encompass aspects of the tumor microenvironment. This could involve developing new classification schemes that incorporate both genetic and pathological signatures, thus providing a more holistic view of CRC biology and potentially refining treatment strategies.

Reviewer #2: This study is clearly structured and logical. Interesting data analyses are presented in the results section and are consistent with the research questions. However, it lacks an effective form of visual presentation. And the protein prognosis studied in the paper needs to be compared with the mRNA prognosis. It would be better if FcFsPs/FcSnPs could be quantified to guide the clinic. Overall, I think your study is of some importance to the field and has potential for publication!

Answer: Thank you for your insightful comments and positive feedback on our study. We acknowledge your suggestion to compare protein prognosis with that of mRNA. The retrospective paraffin-embedded tissues used in this study mostly had degraded mRNA from long storage. This limitation restricts our ability to directly measure mRNA levels. Moreover, the quantification of FcFsPs/FcSnPs to guide the clinical practice is our future study. These were added in the Section of Discussion, yellow highlight.

---

## [Editor Report · Decision Letter 1]

16 May 2024

The combined tumour-based Fascin/Snail and stromal periostin reveals the effective prognosis prediction in colorectal cancer patients

PONE-D-23-41841R1

Dear Dr. Thuwajit,

We’re pleased to inform you that your manuscript has been judged scientifically suitable for publication and will be formally accepted for publication once it meets all outstanding technical requirements.

Kind regards,

Peng Zhang, Ph.D.

Academic Editor

PLOS ONE
---

## [Editor Report · Acceptance letter]

18 Jun 2024

PONE-D-23-41841R1 

PLOS ONE

Dear Dr. Thuwajit, 

I'm pleased to inform you that your manuscript has been deemed suitable for publication in PLOS ONE. Congratulations! Your manuscript is now being handed over to our production team.

Kind regards, 

on behalf of

Prof. Peng Zhang 

Academic Editor

PLOS ONE